# Carbon Redistribution Process in Austempered Ductile Iron (ADI) During Heat Treatment—APT and Synchrotron Diffraction Study

**Xiaohu Li [1,*], Julia N. Wagner [2], Andreas Stark [3], Robert Koos [4], Martin Landesberger [5], Michael Hofmann [4], Guohua Fan [6], Weimin Gan [1] and Winfried Petry [4]**

[1] German Engineering Materials Science Centre (GEMS) at Heinz Maier-Leibnitz Zentrum (MLZ), Helmholtz-Zentrum Geesthacht GmbH, Lichtenbergstr. 1, 85748 Garching, Germany

[2] Institute for Applied Materials (IAM-WK), Karlsruher Institut für Technologie (KIT), Engelbert-Arnold-Straße 4, D 76131 Karlsruhe, Germany

[3] Institute of Materials Research, Helmholtz-Zentrum Geesthacht, Max-Planck-Straße 1, D 21502 Geesthacht, Germany

[4] Heinz Maier-Leibnitz Zentrum (MLZ) FRM II, Technische Universität München, Lichtenbergstr. 1, D 85747 Garching, Germany

[5] Institute of Metal Forming and Casting, Technische Universität München, Walther-Meißner-Str. 4, D 85747 Garching, Germany

[6] School of Materials Science and Engineering Harbin Institute and Technology, Xidazhi Str. 92 Guanlixue yuan 114, 150006 Harbin, China

* Correspondence: xiaohu.li@hzg.de; Tel.: +49-089-289-13982

**Abstract:** In-situ synchrotron diffraction and atom probe tomography (APT) have been used to study the carbon diffusion and redistribution process in austempered ductile iron (ADI) during austempering. The process of carbon content change in bainitic ferrite during different austempering temperatures has been determined quantitatively. The transformation in ADI is controlled by decarburization of supersaturated ferrite and carbide precipitation and has been found to be divided into three stages based on a model developed for bainitic steels by Takahashi and Bhadeshia. The formation, morphology and composition of carbides and carbon clusters in ferrite after austempering have been identified unequivocally by APT. Finally, the relationships of carbon content in ferrite, carbon gap values, and austempering temperatures in the ADI alloy were expressed using empirical equations.

**Keywords:** ADI; synchrotron diffraction; APT; carbon gap; carbide

## 1. Introduction

Austempered ductile iron (ADI) consists of ausferrite and nodular graphite and shows advantageous mechanical properties in many industrial applications [1,2]. To obtain the ausferritic microstructure, the cast ductile iron (DIN 1563) is subjected to a heat treatment involving four steps: (1) austenitization at temperatures between 880–1000 °C, (2) quenching to austempering temperature ($T_{Aus}$, less than 10 s, cooling rate depends on alloying elements), (3) isothermal austempering between 250–450 °C and (4) final cooling down to room temperature. The excellent mechanical properties of ADI are mainly due to this specific ausferrite microstructure, which is determined by the holding (austempering) temperature and time. Ausferrite presents a mixed microstructure of micrometer-sized acicular (or bainitic) ferrite and blocky retained austenite observable in optical micrographs of ADI [3]. The austenite to ferrite transformation (i.e., bainitic transformation) during austempering has a decisive influence on the morphology of ausferrite, and especially, on the bainitic ferrite in ausferrite. Depending

on the different precipitation conditions, bainitic ferrite consists of upper bainite, which is free of carbide precipitates, and lower bainite with fine plate-like carbides in the ferrite grains. After systematically researching this transformation process in steels with different carbon and silicon contents (C = 0.1–1.0 wt%, Si = 0.45–2 wt%) [4–7], a transition temperature between upper and lower bainite around 350 °C has been proposed for a variety of steels by Matas and Hehemann [8]. This transition temperature is a function of the alloying elements and their respective content. Based on their results [8], Takahashi and Bhadeshia [7] developed a model to calculate the transition temperature in plain carbon steels considering the carbon content. This model has been shown to predict accurate transition temperatures of many carbon sheets of steel [7–9].

Taking into account the predictions of this model [7], the bainitic ferrite should; therefore, always consist of lower bainite within the whole austempering temperature range between 250–450 °C in the case of ADI, which has a similar phase and chemical composition as high carbon steel. Lower bainite cannot form in plain carbon steel, which has less than 0.3 wt% C, and upper bainite will disappear in plain carbon steel containing more than 0.4 wt% C [7]. However, the current model gives no explanation on the coexistence of both upper and lower bainite found in certain high carbon steels [9]. Furthermore, it is unable to clarify the absence of carbides in cast iron with a large silicon content at temperatures higher than 350 °C [10,11]. Due to these exceptions and because previous studies were inadequate in describing the carbon redistribution process during bainitic transformation in ADI in detail, it is necessary to precisely measure that process during and after austempering. The results of this paper provide an experimental framework for refining and testing this model, as well as aiming to clarify the influence of unaccounted effects [9–11] that are not included in the present study.

The carbon atoms dissolved in the austenite at high temperature during the first stage of the ADI heat treatment diffuses at the austempering temperature from ferrite into the surrounding austenite matrix. Thus, the retained austenite is stabilized at an austempering temperature so that further transformation ceases [3,12]. In our previous work [3,12,13], the changes in carbon content during austempering was measured using the lattice parameter expansion of ferrite and austenite observed in-situ by neutron diffraction [13]. The number of carbon atoms diffused into the retained austenite matrix was significantly smaller than those diffused out of the ferrite matrix, which was proposed as the carbon gap value in Meier et al. [13].

We used atom probe tomography (APT) and transmission electron microscopy (TEM) to shed light onto the distribution of the excess carbon atoms. Since the time resolution of in-situ neutron diffraction was about 20 s, some important information of carbon diffusion at the beginning of the austempering process was not accessible due to the rapid change of carbon content in ferrite. Therefore, in-situ synchrotron diffraction was used to open up the possibility of investigating previously not examined diffusion processes with high temporal resolution (i.e., one complete diffraction pattern per 2 s).

## 2. Materials and Methods

### 2.1. Preparation of Samples

Plates of ductile iron (GJS) with a dimension of 140 mm × 110 mm × 20 mm cast by Bosch Rexroth AG (Lohr am Main, Germany) were used as the starting material. The chemical composition of the original material is shown in Table 1 (see also [12]).

**Table 1.** Chemical compositions of Austempered ductile iron (ADI) material in weight percent [12].

| C | Si | Ni | Mn | Mo | Mg | Cu | P | S | Ti | Fe |
|------|------|------|------|-------|-------|-------|-------|-------|-------|---------|
| 3.74 | 2.34 | 0.02 | 0.15 | 0.003 | 0.061 | 0.020 | 0.043 | 0.007 | 0.011 | balance |

The base material was then machined to small cylinders (diameter: 6 mm, length: 18 mm) which were then heat-treated in an inert Ar gas atmosphere using a mirror furnace [12,13]. The heat treatment parameters for the different ADI samples [12,13] are listed in Table 2.

**Table 2.** Heat treatment conditions and experimental procedures for all prepared ADI samples. The table also details which measurements were carried out for the respective samples.

| Experiment | Austenitization Temperature [°C] | Austempering Temperature ($T_{Aus}$) [°C] | | | | | | |
|---|---|---|---|---|---|---|---|---|
| | | 250 | 300 | 350 | 375 | 400 | 425 | 450 |
| TEM | 900 | | | × | | | | |
| APT | 900 | | × | | × | | × | |
| In-situ synchrotron diffraction | 900 | × | × | × | | × | | × |

## 2.2. Transmission Electron Microscopy (TEM)

ADI samples austenitized at 900 °C and austempered at 350 °C were mechanically grounded to 100 μm thickness and cut into discs with a diameter of 2 mm. A GATAN691 electro-polishing machine (Gantan, Inc., Pleasanton, CA, USA) was used to thin the disc samples in their center with 4.5 Volt at an angle of 6° to a final thickness of 100 nm. A JEM2100 transmission electron microscope (JEOL Ltd., Tokyo, Japan) was utilized to investigate the morphology of the ausferritic microstructure at room temperature. From the resulting bright/dark field images selected, area diffraction pattern (SADP) analysis of the characteristic spots was used to determine the morphology and orientation relationship between the different phases in ADI.

## 2.3. Atom Probe Tomography (APT)

To quantify the carbon distribution at the atomic level, APT measurements were carried out in voltage mode with a set-point temperature of 50 K, a 50 pJ pulse fraction, a 200 kHz pulse rate, and 0.5% detection rate. Three ADI samples austempered at 300 °C, 375 °C, and 425 °C were prepared by the standard electro-polishing method [14] with further preparation by focused ion beam (FIB). The resulting needle tip-area had a final size of about 250 nm × 60 nm × 60 nm, within which the chemical compositions in different grains and their interface boundaries have been determined. With APT, it was further possible to characterize the spatial structure of phases with grains smaller than 50 nm.

## 2.4. In-Situ Synchrotron Diffraction

In-situ diffraction measurements have been carried out at the beamline P07 (side station) of the PETRA III/DESY synchrotron (Helmholtz-Zentrum Geesthacht GmbH, Hamburg, Germany) [15]. Energy of 87.09 keV was selected yielding a wavelength of 0.14235 Å, and the sample to detector distance was set to 1558.6 mm. The beam size was 0.7 mm × 0.7 mm, and the effective gauge volume was about 2 mm$^3$. The measurements were used to determine the carbon diffusion between austenite and ferrite during the austempering treatment and the carbon content in ferrite after austempering via determination of the lattice parameters. Due to the high quenching rates, as well as the required temperature precision and stability, a quenching dilatometer (TA DIL805, TA instruments, New Castle, DE, USA) with inductive heating was used for the in-situ ADI heat treatment for all the synchrotron measurements (Figure 1). After quenching, the sample temperature reached the austempering holding temperature within a time of less than 3 s with a maximal temperature fluctuation of 0.1 °C.

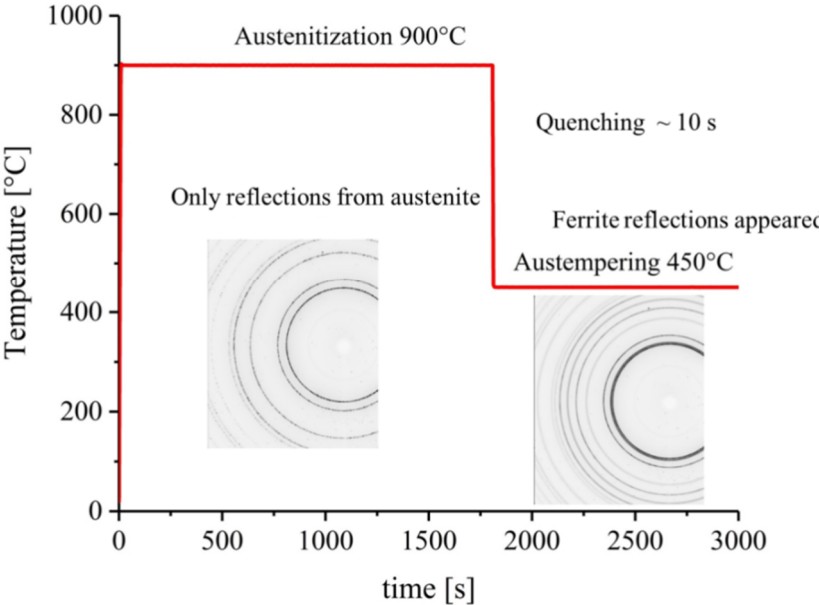

**Figure 1.** A typical temperature curve visualizing the high-temperature stability of the dilatometer during the in-situ experiment of austempered ductile iron (ADI) is shown exemplarily for an austempering temperature of $T_{Aus}$ = 450 °C. Also shown as an example are diffraction patterns obtained at $T$ = 900 °C (only austenite) and during austempering at $T_{Aus}$ = 450 °C, respectively.

For this, the respective cylindrical ADI sample (length 10 mm, diameter 4 mm) was spot welded with a type S thermocouple on its surface and placed in the center of the heating coil of the dilatometer. The samples were held in place by two alumina rods. The heat treatment parameters and measurement times are listed in Table 3.

**Table 3.** Heat treatment conditions of ADI samples used in the in-situ synchrotron diffraction experiments. The duration time of the fast measurement mode at the start of austempering is also given. The austempering time at different temperatures was chosen based on previous work [3] to make sure that the austempering stopped within the ADI processing window (i.e., both carbon diffusion and phase transformation have been completed).

|  | Austempering $T_{Aus}$ [°C] | Time [min] | Time (in [min]) at Fast Mode with Time Resolution of 2 s |
|---|---|---|---|
| **ADI (0 wt% Ni)** | 450 | 30 | 5 |
|  | 400 | 40 | 10 |
|  | 350 | 60 | 20 |
|  | 300 | 90 | 30 |
|  | 250 | 120 | 45 |

*2.5. Calculation of Carbon Content in Ferrite*

The calculation of carbon content in ferrite is based on the linear relationship between carbon content and the ferrite lattice parameter. The specific volume of ferrite increases with increasing temperature continuously due to the thermal vibration of iron atoms. The associated temperature dependence of the ferrite lattice parameter in pure iron was measured using X-ray diffraction [16–20] and summarized in [3,12]. Taking into account the influence of silicon as a substitutional element in ferrite, the lattice parameter of ferrite with 0 wt% carbon could be expressed in Equation (1) in the temperature range from 20 °C to 750 °C.

$$a_{\alpha}^{Ref}(T) = a_{\alpha,RT}^{Ref} \cdot \left(1 + \alpha_{\alpha}^{l} \cdot T + \alpha_{\alpha}^{q} \cdot T^2\right) + a_{Si}^{l} \tag{1}$$

$a_\alpha^{Ref}(T)$: Lattice parameter of ferrite at temperature $T$ [Å]

$a_{\alpha,RT}^{Ref}$: Lattice parameter of ferrite at room temperature (20 °C), 2.8663 Å

$\alpha_\alpha^l$: Linear thermal expansion coefficient, $1.294 \times 10^{-5}$ K$^{-1}$

$\alpha_\alpha^q$: Square of the thermal expansion coefficient, $2.729 \times 10^{-9}$ K$^{-2}$

$T$: Temperature difference to ambient temperature [K]

$a_{Si}^l$: Lattice expansion of ferrite due to silicon, 2.5 wt% Si in ADI leads to $-0.003$Å expansion.

The additional lattice expansion of ferrite by carbon in Fe-C alloys up to the maximum carbon solubility ($x_\alpha^c = 0.02$ wt%) has also been measured precisely and can be expressed by Equation (2) [21]. Takahashi et al. have proposed that carbon atoms in supersaturated ferrite are stored in octahedral as well as tetrahedral vacancies within the ferrite lattice [17]. The additional occupancy of carbon atoms in tetrahedral vacancies would lead to a reduced theoretical lattice expansion coefficient of $k_\alpha^c = 0.0079$ Å/wt% instead of 0.0385 Å/wt% (which is the experimentally observed value for $x_\alpha^c \leq 0.02$ wt%). However, these theoretical calculations have not been supported or falsified by any accurate experimental data to the best of our knowledge. Therefore, the calculations of carbon content in supersaturated ferrite in this work are based solely on Equation (2) using $k_\alpha^c = 0.0385$ Å/wt%. In addition, stress caused by the different expansion coefficients of austenite and ferrite during the formation could also influence the lattice parameter. However, Hall et al. [22] as well as Meier [3] showed that the resulting strain during the formation of ferrite in the austenite matrix gives only a comparatively minor contribution to the lattice change and consequently is not included in our calculations.

$$a_\alpha^c = a_\alpha^0 + k_\alpha^c \cdot x_\alpha^c \tag{2}$$

$a_\alpha^c$: Lattice parameter of ferrite

$a_\alpha^0 = a_{\alpha,RT}^{Ref}$: Lattice parameter of ferrite by $x_\alpha^c = 0.0$ wt% [Å]

$k_\alpha^c$: Lattice expansion coefficient of ferrite due to carbon atoms, 0.0385 Å/wt%

$x_\alpha^c$: Carbon content in ferrite [wt%]

## 3. Results and Discussion

### 3.1. Ausferritic Microstructure

In former research work on steels which contain both ferrite and austenite phases [23–25], bright field (BF) images of ferrite exhibit a bright colour while austenite appears as a dark colour due to their different crystal structures and carbon contents [26]. A typical BF-TEM image and corresponding SADP is shown in Figure 2a,b for an ausferritic microstructure resulting from an austempering treatment at 350 °C. This microstructure is quite similar to morphologies found in typical bainitic steels. The austenite grains occur mainly in needle-mesh form and are scarcely found as massive blocks (Figure 2c), while the shape of ferrite as seen in Figure 2a–c is mostly massive in appearance [27]. The ausferrite appearance in TEM is quite different from the one found typically with optical microscopy. The typical ausferrite microstructure consisting of needle-like ferrite and blocky austenite observed with optical microscopy cannot be found in TEM. This is attributed to the limited magnification of the optical microscopy as compared to the high-resolution TEM images. According to the dark field image using a Bragg reflection of austenite (Figure 2c), the length of the austenite needles can be estimated at about 100–500 nm with a width of 20–100 nm, while ferrite grains are found in sizes of about 50–300 nm. Additionally, in the SADP, a typical crystallographic orientation relationship ([001]F//[01$\bar{1}$]A), labeled as Nishiyama–Wasserman (N-W)) between austenite and ferrite has been found and is depicted in Figure 2b. This also is in good agreement with recent texture measurements showing a similar orientation relationship in bulk samples [28,29].

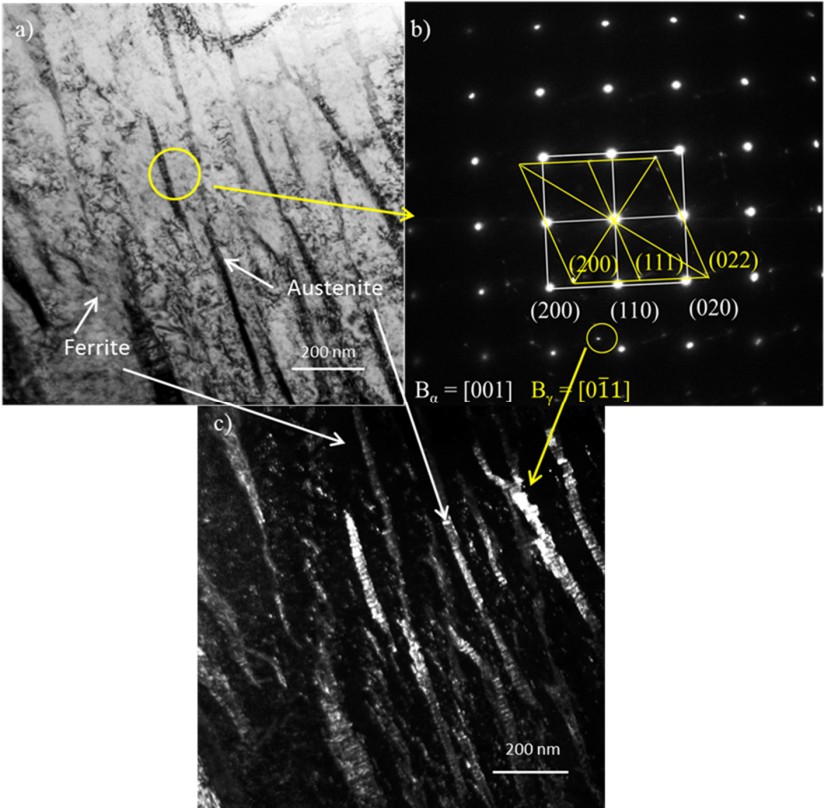

**Figure 2.** (**a**) The bright-field (BF) image of ADI austempered at 350 °C for 30 min. (**b**) The associated SADP was taken from the area marked by the yellow circle in (**a**). The orientation relationship between austenite (yellow) and ferrite (white) is also shown, as discussed in the text. (**c**) Dark-field image of (**a**) using the Bragg spot highlighted by a circle in (**b**).

A small amount of carbide precipitation has been observed nearby the austenite grains (Figure 3a). Four types of stable carbides ($\theta$-$Fe_3C$, $Fe_6C$, h-$Fe_7C_3$, and $\gamma$-$Fe_{23}C_6$) are most likely to form in cast iron [30,31]. From the orientation of the Bragg scattering spots of carbide in the SADP (Figure 3b), the crystal structure of the carbides can be most likely identified as $\gamma$-$Fe_{23}C_6$. The formation of carbides is consistent with the existence of the carbon gap in ADI. Formation of carbides, their location, and chemical composition will be discussed in detail in the following section.

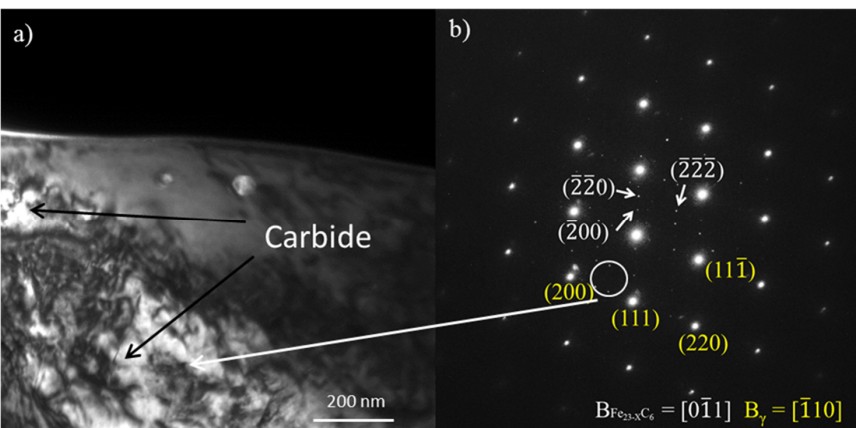

**Figure 3.** (**a**) Dark-field image of ADI austempered at 350 °C for 30 min. (**b**) The SADP shown, is taken from the complete area in (**a**).

### 3.2. Carbon Diffusion Process and Carbon Content in Ferrite and Austenite

### 3.2.1. Synchrotron Diffraction

In our previous in-situ neutron diffraction studies on carbon diffusion in ADI, the carbon content changes in austenite and ferrite during austempering have been systematically investigated [3,12,13]. However, the neutron diffraction setup used in these experiments had some limitations for in-situ measurement. These were the coarse time resolution of $t \geq 20$ s, the small detector coverage in scattering angle $2\theta$ restricting the phase analysis to just three Bragg reflections [3,13] and the larger instrumental peak broadening compared to synchrotron diffraction. Hence, the carbon content in ferrite could not be determined accurately in those studies with volume fractions lower than 2 vol%. Therefore, at the start of the second stage of the heat treatment process—e.g., the first 2–5 mins depending on the austempering temperature [3,13]—only very limited results on ferrite formation could be obtained. Consequently, we used synchrotron diffraction with a high temporal resolution to extract information of the emerging ferrite phases during that early period.

Ferrite lattice constants were derived from fits to the peak profiles of the synchrotron diffraction patterns using the program STECA [32] and are shown as a function of time for different austempering temperatures in Figure 4a. In Figure 4, $t = 0$ s is set as the time when the sample temperature first reaches the corresponding austempering temperature.

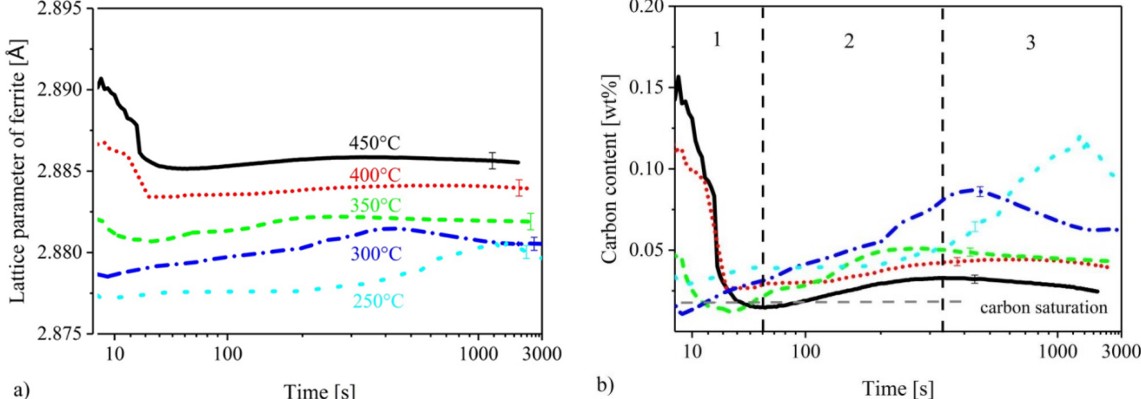

**Figure 4.** (**a**) Evolution of the lattice parameters of ferrite in ADI during austempering at a different temperature between 250–450 °C. $t = 0$ s is the point in time when the temperature just reached the austempering temperature. (**b**) Carbon content change as converted from the lattice parameter according to Equation (2). Exemplarily for the case of ADI austempered at 450 °C, the vertical lines depict the three stages of the carbon diffusion process regarding decarburization and carbide precipitation, as explained in the text. For the sake of clarity, the errors are shown only at a single point for each curve.

Based on the introduction about the ferrite lattice parameter in Section 2.5 and other publications [6,33], it can be confirmed that the substitutional atoms in the iron matrix-like manganese and silicon will not diffuse during austempering. During austempering, changes in the lattice parameter of ferrite are from the diffusion of carbon, only. Equations (1) and (2) were used to convert the lattice parameter of ferrite into carbon content. This allows depicting the evolution of carbon with austempering time excluding the interference of lattice expansion with temperature and of the alloying silicon. The resulting graph is shown in Figure 4b.

The bainitic transformation already starts during the quenching process after less than 10 s (quench rate = 100 K/s). The ferrite (110) reflection appears first at temperatures between 700–650 °C, which could not be detected in the comparably slow in-situ neutron diffraction experiments [3] and it is also different to the calculations of the Time Temperature Transformation (TTT) diagram of ADI [34]. Here we find that the formation of ausferrite starts before the 1% ausferrite curve shown in the TTT diagram,

highlighting the extra insight the synchrotron measurements provided during the early transformation stages of the austempering process.

The lattice parameter (i.e., carbon content) of ferrite as a function of time in all the investigated samples initially shows a decrease, then an increase, followed by a subsequent decrease again (Figure 4b). Exemplarily in Figure 4b, the evolution of carbon content in ferrite at 450 °C could be divided into three stages considering decarburization of supersaturated ferrite, precipitation of carbides, and the bainitic phase transformation rate (Note: In-situ neutron diffraction is too slow to visualize the 1st stage and the start of the 2nd stage. Only part of the information of the 2nd stage and the complete 3rd stage are accessible by neutron diffraction [3,12].)

Stage 1 (duration time $t < 60$ s): The decrease of carbon content in ferrite could be clearly observed in ADI during austempering at 350 °C, 400 °C, and 450 °C. That process appears to be too fast during austempering at 300 °C and 250 °C (Figure 4a) to be unequivocally trackable with diffraction techniques. Thus, ADI materials austempered at 300 °C and 250 °C can only be observed at stage 2 and 3, as depicted in Figure 4. Hence, almost no carbon diffusion from ferrite into retained austenite could be found at the start of austempering process at temperatures below 300 °C. The explanation for this behaviour is, that according to the Takahashi and Bhadeshia model [7] for temperatures lower than 300 °C, the bainitic transformation is controlled by the carbide precipitation only ($t_C < t_d$, $t_C$ is in the range of 0.1–1 s, $t_d$ is between 1–10 s). ($t_C$ is the time of carbide precipitation, $t_d$ is the time of decarburization of supersaturated ferrite.)

Consistent with this model, the APT result in Figure 5a shows the frequent occurrence of carbides at the ferrite/austenite interface for ADI austempered at 300 °C. Both carbide precipitation and carbon diffusion rates are generally higher for austempering temperatures above 350 °C, but the difference between $t_C$ and $t_d$ is considerably smaller ($t_C < 0.1$ s, $t_d$ is between 0.1–1 s). This indicates that the bainitic transformation is no longer dominated by carbide precipitation when the austempering temperature exceeds 350 °C, and carbon diffusion will become more prevalent. Thus, the change of carbon content in ferrite at higher holding temperatures takes longer.

Stage 2 (duration time 180 s $< t <$ 20 min): The duration of this stage depends on austempering temperature and lasts from approximately 3 min (450 °C) to 20 min (250 °C), respectively. Our previous work indicates, that the phase transformation rate is the highest at this stage, and the carbon content with retained austenite also increases from 0.8 wt% to about 90% of its maximum at the end of this stage (1.55 wt% to 1.7 wt%, depending on the austempering temperature) [3,12]. The increase of the average ferrite lattice parameter in this stage is similar to the early stages of bainitic transformations in high silicon steel [35]. As the bainitic transformation continues, the carbon content between ferrite and austenite will undergo a process from non-equilibrium to an equilibrium state. This process occurs at the ferrite/austenite interface, simultaneously with ferrite growth during bainitic transformation. In that, the ferrite growth is accompanied and controlled by macroscopic partitioning of carbon between ferrite and austenite [4]. Several models describing the carbon content equilibrium at the ferrite/austenite interface exist [4]. The driving force of carbon diffusion from ferrite to austenite will decrease as the carbon content in austenite increases; as a consequence, the carbon content in newly formed ferrite raises subsequently.

Stage 3: To the best of our knowledge, the small decrease of the ferrite lattice parameter during austempering in this stage has not been observed using in-situ neutron diffraction or any other in-situ techniques before. This is related to the measurement accuracy of the in-situ neutron diffraction technique and will be discussed in detail in Section 3.3. With the phase transformation rate continuing to decrease, a large amount of austenite has already transformed into nano-size ferrite. However, due to the continuous increase of carbon content in austenite during this stage [3], the driving force of further carbon diffusion is not large enough. As constrained para-equilibrium, as well as negligible partitioning local equilibrium (NPLE), like in quench & partitioning steel (Q&P) or steels with high carbon and silicon content cannot be achieved in unalloyed ADI [4,35,36], the carbon content in ferrite is always maintained in a supersaturated state.

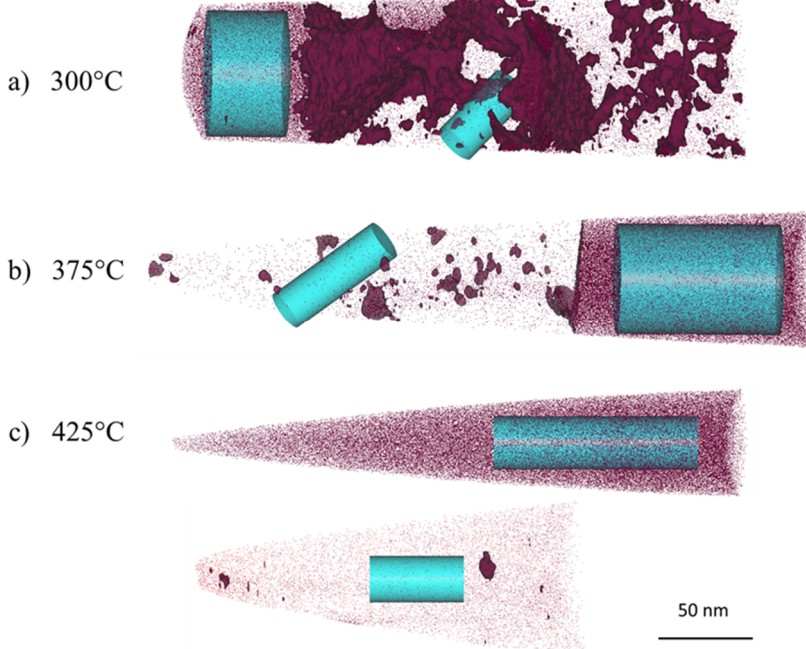

**Figure 5.** APT maps of carbon atoms in ADI (from [37]). Red dots: carbon atom. Cyan cylinder: the area from which the averaged carbon content was calculated. (**a**) ADI austempered at 300 °C for 60 min. (**b**) ADI austempered at 375 °C for 43 min. (**c**) ADI austempered at 425 °C for 13 min.

### 3.2.2. Atom Probe Tomography (APT)

Three ADI samples austempered at different temperatures (300 °C, 375 °C, and 425 °C) were measured using APT to study the resulting carbon distribution in austenite and ferrite phases. The austempering time was chosen according to our earlier work as the time when the phase transformation rate drops to $10^{-6}$ vol%/s$^{-1}$ for each corresponding temperature [3,12,13].

Here it was found that a considerable fraction of carbon dissolved during austenitization was not redistributed into the retained austenite at the end of the heat treatment. This leads to the identification of the so-called "carbon gap" defined by [3]. It was found that at the lowest austempering temperatures (i.e., 250 °C), the carbon gap reached the highest level, while it was minimal at around 425 °C [3,12].

Figure 5. shows the distribution of carbon atoms in the three different samples as measured by APT. This allows to qualitatively compare the samples. From the higher density of carbon atoms, austenite grains could clearly be identified. In austenite, we find a quite uniform distribution of carbon, while in the ferrite grains of all the investigated ADI samples, partial segregation of carbon atoms could be identified. In all samples, the respective chemical compositions were extracted from selected areas within the prepared needle-shaped samples (cyan cylinders) with the results listed in Table 4. As noted above, Si and Mn show an almost uniform distribution throughout the samples, irrespective of the holding temperature.

**Table 4.** Averaged chemical composition (at%) in the cylinder (Figure 5), errors on chemical compositions are given in brackets.

| Element | ADI (300 °C for 60 min) | | ADI (375 °C for 43 min) | | ADI (425 °C for 13 min) | |
|---|---|---|---|---|---|---|
| | **Austenite** | **Ferrite** | **Austenite** | **Ferrite** | **Austenite** | **Ferrite** |
| Fe | 88.6(2) | 94.4(3) | 88.36(1) | 95.24(6) | 89.73(1) | 95.0(3) |
| C | 6.38(1) | 0.30(1) | 7.27(1) | 0.17(1) | 5.46 (1) | 0.10(1) |
| Si | 4.68(1) | 5.1(3) | 3.896(6) | 4.2(1) | 4.37(1) | 4.6(3) |
| Mn, Cr, Al, O, etc. | 0.34(1) | 0.20(1) | 0.48(1) | 0.40(1) | 0.44(1) | 0.30(1) |

Figure 6 compares the carbon content in ferrite as determined by synchrotron diffraction and APT versus austempering temperature (Figure 4 and Table 4). Other than synchrotron diffraction, which is essentially a bulk method, APT provides very localized information on the chemical content. However, Figure 6 clearly shows a good agreement between both methods. The results indicate a direct relationship between carbon content in ferrite and the austempering temperature. This can be expressed through a simple linear extrapolation (Equation (3)) for the temperature range $T_{Aus}$ = 250–450 °C. This relationship indicates that the ferritic phase is supersaturated with carbon throughout the investigated temperature region. The maximal carbon content is found for low austempering temperature consistent with the highest carbon gap values at those temperatures [3,12] and the model developed by Takahashi and Bhadeshia [7].

$$x_C^{ferrite} = -0.0014 \cdot T_{Aus} + 0.7 \tag{3}$$

$x_C^{ferrite}$: Carbon content in ferrite [at%]

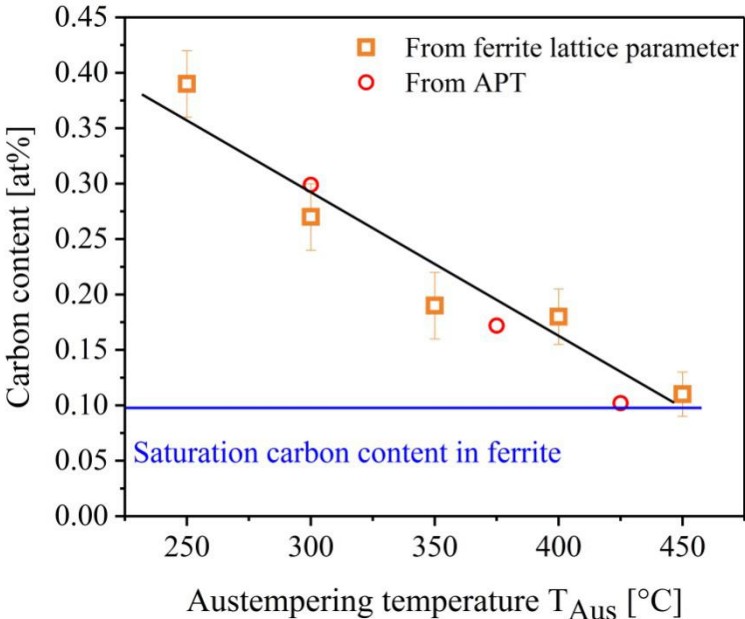

**Figure 6.** Carbon content [at%] in ferrite vs. austempering temperature $T_{Aus}$. Orange squares: results after calculation using synchrotron diffraction data (Figure 4). Red circles: from the APT measurements (Table 4).

From the carbon distribution derived from the 3-D atom maps, two cases could be identified on how carbon segregates in ferrite. Close to the austenite/ferrite interface, carbon appears mainly in sheet-like or blocky carbon-enriched areas with a typical carbon content ≥20 at% (Figures 5a and 7a), while inside the ferrite grains sporadic small granular areas having a carbon content between 5–15 at% are found (Figures 5b and 7b). From the chemical elements in the carbon segregation area, the results of the SADP measurements (cf. Figure 3) and earlier research work on iron-based materials [4,27,30,33] that undergo a similar phase transformation, one can identify the sheet-like or block carbon regions most probably as θ-$Fe_3C$ and γ-$Fe_{23}C_6$. On the other hand, the ratio between carbon and iron atoms in the smaller granular parts does not coincide with any known carbide composition. For this, areas with the most likely explanation are granular retained austenite (carbon content ≤ 8 at%) and carbon clusters (Carbon content 8–15 at%) [37].

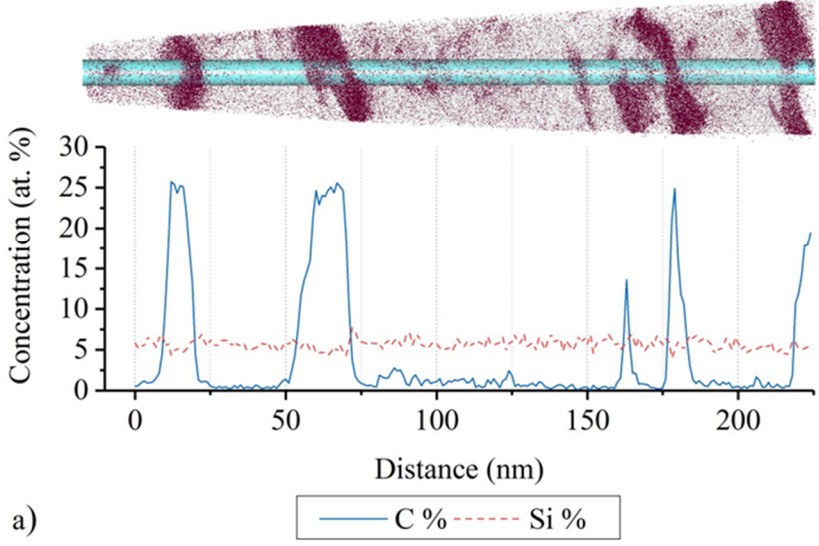

a)

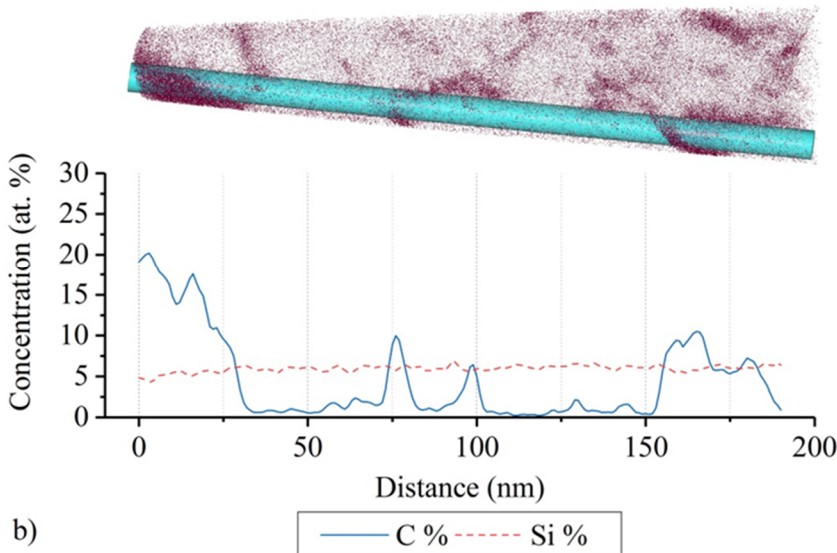

b)

**Figure 7.** (**a**) APT map of ADI austempered at 300 °C for 60 min. The silicon Si and carbon C concentration profiles in the selected area are plotted directly below. (**b**) Another APT map of the same sample with the silicon Si and carbon C concentration profiles showing directly below.

However, as stated above APT is a local probe, and therefore, the content and morphology of carbide formed cannot easily and directly be compared to the bulk averaged results from diffraction experiments. The temperature range of carbide precipitation in Fe-based alloys is greatly affected by the chemical compositions and heat-treatment conditions [27], there are currently no empirical equations to describe their relationship. However, the carbide forms in ferrite through the entire austempering temperature range (250–450 °C), which is again consistent with the results predicted by the model of Takahashi and Bhadeshia [7].

### 3.3. Carbon Gap

Combining all the results from the TEM, synchrotron diffraction, as well as APT, the unaccounted carbon (or the carbon gap) in ADI materials [3] could be clarified. Initially, the carbon content in austenite is about 0.8 wt% ($\omega_C^{matrix}$) after austenitization and before the holding heat treatment as shown by the synchrotron and neutron diffraction consistent also with the Fe-C-Si phase diagram [3].

To calculate the carbon gap ($\omega_{C-gap}$) for different austempering temperatures, first, the retained austenite phase fraction was determined by Rietveld analysis of the diffraction data. Then the carbon content of the retained austenite ($\omega_C^{austenite}$) was derived from the lattice expansion [38] and substituted into Equation (4). The results of this calculation are shown in Figure 8. For comparison, the $\omega_C^{ferrite}$ in Equation (5) has been calculated from the average carbon content of ferrite (from Figure 6) and the fraction of ferrite determined by Rietveld analysis and is also shown in Figure 8. The results show that only about 9% to 12% of excess carbon atoms (from the carbon gap) in the ferrite matrix form supersaturated ferrite. The remaining carbon atoms are distributed as carbides and carbon clusters in the ferritic grains

$$\omega_{C-gap} = \omega_C^{matrix} - \omega_C^{austenite} \tag{4}$$

$$\omega_C^{carbide+carbon\ cluster} = \omega_{C-gap} - \omega_C^{ferrite} \tag{5}$$

$\omega_C^{matrix}$: Measured carbon content in austenite matrix at 900 °C [wt%]

$\omega_{C-gap}$: Carbon gap in ADI [wt%]

$\omega_C^{austenite}$: Carbon content in the austenite matrix [wt%]

$\omega_C^{carbide+carbon\ cluster}$: Carbon content in carbide and the carbon cluster [wt%]

$\omega_C^{ferrite}$: Carbon content in the ferrite matrix [wt%]

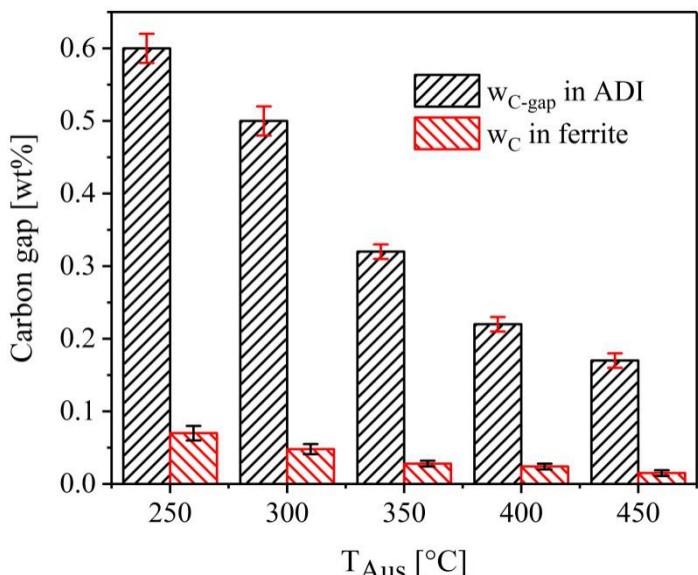

**Figure 8.** Carbon gap values $\left(\omega_{C-gap}\right)$ and the dissolved carbon content ($\omega_C$) in ferrite in ADI austempered at different temperatures.

The values for the carbon gap (Figure 8) derived from the synchrotron diffraction data are in excellent agreement to the values found previously using neutron diffraction [3,12]. The difference between the calculated carbon gap values using these two different methods is less than 0.05 wt%. Obviously, both diffraction methods could be used to derive accurate values on carbon distribution as a function of heat-treatment temperatures. However, the lattice parameter of ferrite obtained by in-situ synchrotron diffraction measurements was calculated from an entire synchrotron diffraction pattern (data for diffraction pattern were integrated over 270° of the complete 360° Debye–Scherrer rings) with accurate wavelength calibration. Generally, this yields higher precision of the values compared to the in-situ neutron diffraction results, which were based on the observation of single Bragg reflections. This is clearly seen in the lattice parameter expansion of ferrite at the end stage of austempering. Essentially, here the carbon diffusion and phase transformation rate has ceased

and reached their respective plateau values. The carbon content in ferrite determined by single peak neutron diffraction data has larger errors as compared to the whole pattern analysis results from the synchrotron experiments [3]. Therefore, the carbon content in ferrite could not unequivocally be calculated from the carbon gap values using only neutron diffraction data [3]. Nevertheless, the small amount of carbon dissolved in the ferritic matrix (9–12%, Figure 8) does not modify the previously observed trend between carbon gap values and austempering temperature. The spatial distribution of carbon atoms indicates that all unaccounted carbon atoms are solely contained within the ferrite matrix either directly dissolved (10%) within or precipitated (~90%) as carbides or clusters. In addition, Figure 8 also confirms that the carbon content dissolved in the ferrite matrix follows the same trend as the total carbon gap, which reduces continuously with increasing austempering temperature. This is due to the increasing diffusion rate of carbon at a higher temperature, which in turn enables faster redistribution into either carbides or the austenite matrix.

## 4. Conclusions

The main results and new findings of the current investigation on austempered ductile iron can be summarized as follows:

- Using the relationship between the ferrite lattice parameter and the carbon content in ferrite, the whole process of carbon diffusion between austenite and ferrite during austempering was investigated by in-situ synchrotron diffraction. The ccontent changes in ferrite during austempering have been clarified to be related to two processes, decarburization and carbide/carbon cluster formation.
- In-situ synchrotron diffraction data and APT measurements indicate the formation of supersaturated ferrite in ADI. The amount of carbon dissolved in the ferrite matrix follows a linearly decreasing trend as the austempering temperature increases. The ferrite was found to be in a carbon-supersaturated state within the full range of austempering temperatures for all ADI materials investigated.
- Carbon redistribution in austenite and ferrite after austempering has been shown directly by APT. The carbon gap values in ADI are found to be inversely proportional to the holding temperature. Of all unaccounted carbon atoms constituting the carbon gap about 90% of carbon atoms are present as carbides and carbon clusters with only 10% dissolved in the ferrite matrix.

**Author Contributions:** Conceptualization—X.L.; Methodology—this work is part of the PhD thesis of X.L. [39], X.L., J.N.W., A.S. and G.F.; Data curation—X.L. and M.L.; Writing—original draft preparation—X.L.; Writing—review and editing—X.L., W.G., R.K., and M.H.; Project administration—M.H. and W.P.; Funding acquisition—M.H. and W.P.

**Funding:** Funding by Grant PE 580/14-1 (Project: 233737539) and PE 580/16-1 (Project: 28975656) of the Deutsche Forschungsgemeinschaft (DFG) is gratefully acknowledged.

**Conflicts of Interest:** The authors declare no conflict of interest.

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
