# Peer review of "Carbon Redistribution Process in Austempered Ductile Iron (ADI) During Heat Treatment—APT and Synchrotron Diffraction Study"

_metals, doi:10.3390/met9070789_

Round 1
Reviewer 1 Report
The work is of good quality and contributes to the understanding of the subject matter. The manuscript could be accepted for publication without any further revision. Also, in the conclusion, it would be good to emphasize what are new findings of this research.

Author Response
The work is of good quality and contributes to the understanding of the subject matter. The manuscript could be accepted for publication without any further revision. Also, in the conclusion, it would be good to emphasize what are new findings of this research.
We thank the referee for his generous remarks on our paper. All misprints which were highlighted by the referee were corrected. Furthermore we added the following sentence to highlight our new findings in the conclusion, as suggested.
“The main results and new findings of the current investigation on austempered ductile iron can be summarized as follows:”
Reviewer 2 Report
The manuscript is well written: the theoretical background is clear, the experimental techniques and the testing matrix are clearly reported, as well as the results are well analysed and rationalised.
There are several misprints to be corrected.

Author Response
The manuscript is well written: the theoretical background is clear, the experimental techniques and the testing matrix are clearly reported, as well as the results are well analysed and rationalised.
There are several misprints to be corrected.
We thank the referee for his generous remarks on our paper. All misprints were corrected and amended. We improved the contrast level in Figure 3 to make the small Bragg spots more visible.
Reviewer 3 Report
Excellent paper, the set of unique experimental methods was used and the interesting results were delivered.
P.S. I have a question for the authors about, so called, carbon cap, or “not mass balanced” carbon before quenching and after austempering, Where these “hidden” carbon atoms locate in such numbers that even atomic probe tomography cannot count it? Let me suggest to consider two possible (hypothetical) mechanisms which could be involved: (i) fast diffusion of the part of carbon atoms to exiting graphite nodules during not momentarily quenching. In SGI, nearest neighboring distance in a range of less than hundred microns, considering nodule diameter, the actual carbon diffusion distance will only few tens microns; this is a difference between SGI and steels which do not have pure graphite phase; (ii) we can predict that a large internal interface stress takes place during austempering because of a difference in FCC/BCC volumes; this stress will develop strain and influences on the measured lattice parameters in addition to carbon concentration. So, the real carbon concentration in the phases of austempered SGI could be differ from calculated with used in article Eq. which considered only the effect of alloying elements.
To address my comments in this article is up to the authors.
Author Response
Excellent paper, the set of unique experimental methods was used and the interesting results were delivered.
P.S. I have a question for the authors about, so called, carbon cap, or “not mass balanced” carbon before quenching and after austempering, Where these “hidden” carbon atoms locate in such numbers that even atomic probe tomography cannot count it? Let me suggest to consider two possible (hypothetical) mechanisms which could be involved: (i) fast diffusion of the part of carbon atoms to exiting graphite nodules during not momentarily quenching. In SGI, nearest neighboring distance in a range of less than hundred microns, considering nodule diameter, the actual carbon diffusion distance will only few tens microns; this is a difference between SGI and steels which do not have pure graphite phase; (ii) we can predict that a large internal interface stress takes place during austempering because of a difference in FCC/BCC volumes; this stress will develop strain and influences on the measured lattice parameters in addition to carbon concentration. So, the real carbon concentration in the phases of austempered SGI could be differ from calculated with used in article Eq. which considered only the effect of alloying elements.
To address my comments in this article is up to the authors.
Many thanks for the very helpful comments. We would like to comment as follows:
The carbon of the “gap” is mainly found in the carbides and small clusters formed in the ferrite matrix during austempering. Indeed the carbon diffusion is very short ranged with only a few microns, together with the very small grain sizes we believe this to be the reason for the formation of these clusters and carbides where more than 90% of all carbon gap atoms reside at the end of the transformation. This clusters and carbides could be clearly observed by us in the APT as we think we have shown in the current paper (we also will elaborate this topics and APT measurements in further publication which will be submitted in the very near future).
The second point concerning interphase stress is clearly valid and we included the following small paragraph addressing this in the section 2.5 (“Calculation of carbon content in ferrite”) and added further reference.
“In addition stress caused by the different expansion coefficients of austenite and ferrite during the formation could also influence the lattice parameter. However, Hall et al [22] as well as Meier [3] showed that the resulting strain during formation of ferrite in the austenite matrix gives only a comparable minor contribution to the lattice change and consequently is not included in our calculations.”